# A Dual-Purpose Camera for Attitude Determination and Resident Space Object Detection on a Stratospheric Balloon

**DOI:** 10.3390/s24010071

**Published:** 2023-12-22

**Authors:** Gabriel Chianelli, Perushan Kunalakantha, Marissa Myhre, Regina S. K. Lee

**Affiliations:** Department of Earth and Space Science and Engineering, York University, Toronto, ON M3J 1P3, Canada; pk7@my.yorku.ca (P.K.); marissarmyhre@gmail.com (M.M.); reginal@yorku.ca (R.S.K.L.)

**Keywords:** Space Situational Awareness (SSA), star tracker, Resident Space Objects (RSOs), attitude determination, object detection, stratospheric balloon, Commercial off-the-shelf (COTS)

## Abstract

Space systems play an integral role in every facet of our daily lives, including national security, communications, and resource management. Therefore, it is critical to protect our valuable assets in space and build resiliency in the space environment. In recent years, we have developed a novel approach to Space Situational Awareness (SSA), in the form of a low-resolution, Wide Field-of-View (WFOV) camera payload for attitude determination and Resident Space Object (RSO) detection. Detection is the first step in tracking, identification, and characterization of RSOs, including natural and artificial objects orbiting the Earth. A space-based dual-purpose camera that can provide attitude information alongside RSO detection can enhance the current SSA technologies which rely on ground infrastructure. A CubeSat form factor payload with real-time attitude determination and RSO detection algorithms was developed and flown onboard the CSA/CNES stratospheric balloon platform in August 2023. Sub-degree pointing information and multiple RSO detections were demonstrated during operation, with opportunities for improvement discussed. This paper outlines the hardware and software architecture, system design methodology, on-ground testing, and in-flight results of the dual-purpose camera payload.

## 1. Introduction

Space Situational Awareness (SSA) refers to the ability to detect, track, identify and characterize Resident Space Objects (RSOs), and is crucial in understanding and managing the environment in the Earth’s orbit. It involves the mitigation of space debris, collision avoidance, protection of space assets, Space Traffic Management (STM), and supports scientific research and national security. We have already experienced devastating impacts resulting in the loss of telecommunication and creation of debris in the Earth’s orbit in the 2009 collision of Iridium 33 and Kosmos 2251 [1]. As the number of satellites increases in the coming years, the probability of similar collisions is likely to occur every five to nine years [2]. With this, the need for robust and resilient SSA systems becomes more pressing. The Space Surveillance Network tracks tens of thousands of objects larger than 10 cm in diameter. However, there are hundreds of thousands of smaller debris that are considered too small to track or catalogue. This places importance on the continual development of SSA infrastructure to address the concerns of detection and tracking of RSOs using novel and advanced technologies.

A low-cost, Wide Field-of-View (WFOV) dual-purpose camera system can serve as both an attitude sensor and SSA payload on a nanosatellite platform. Space-based observations provide a unique opportunity to support ground infrastructure with the benefit of improved imaging conditions due to the lack of atmospheric interference and longer access times in orbit. Star trackers (ST), commercial-grade camera systems with space-flight heritage, are similar in technology and perform attitude determination (AD). Integration of RSO detection can enhance current space technology and merge spacecraft attitude information with SSA. In 2022, we demonstrated a ST-like camera system for Space-Based Space Surveillance (SBSS) on a stratospheric balloon in collaboration with the Canadian Space Agency (CSA) and the National Centre for Space Studies (CNES) (Figure 1). More details of the STRATOS 2022 mission can be found in [3]. We validated the performance of this new star tracker design through simulation and in situ ground measurements, and continue to develop image processing algorithms, Field-Programmable Gate Array (FPGA) camera electronics and mission concept studies for future SSA missions.

In this paper, we present the improved dual-purpose camera system, demonstrated on the 2023 STRATOS Balloon platform with real-time operation of a star tracker with simultaneous RSO detection. The 2023 campaign was to demonstrate real-time attitude determination and RSO detection and identify the challenges with the implementation of the dual-purpose star tracker.

### 1.1. Dual-Purpose Star Tracker for RSO Detection

The concept of a dual-purpose star tracker is not new. In [4], our team demonstrated that STs are cost-effective, flight proven, and require basic image processing to be used as an attitude-determination sensor. An AI-based RSO detection algorithm was presented in this paper “to augment the capabilities of a star tracker by becoming an opportunistic space-surveillance sensor”. In [5], we also examined the feasibility of a virtual constellation using dual-purpose star trackers for Space Domain Awareness (SDA) and applications. Similar concepts were also examined in [6,7,8]. While these articles present a similar concept of using star trackers for detecting RSOs in star field images, there has been little to no in-orbit demonstration of the multi-use star trackers with real-time performance of the functions. Dual-purpose star trackers for in-orbit tests and operation were presented in [9,10], respectively. Both aim to demonstrate and evaluate the performance and functional requirements of a star tracker as a SSA sensor. The work presented in this paper is aimed to develop the technology in a stratospheric balloon platform with the eventual space-based nanosatellite application.

As discussed in the references listed above, dual-purpose star trackers present a low-cost, yet effective alternative to dedicated space surveillance payloads. Most SSA missions rely on expensive imaging systems equipped with telescopes to provide high resolution RSO images. Instead, star trackers can serve as a secondary SSA payload or proximity sensor to continuously monitor the host satellites surroundings. As Low Earth Orbit (LEO) continues to become more congested and contested, proximity sensing continues to be recognized as a tool to avoid in-orbit collisions, raise awareness of the surroundings, and provide SSA data when necessary. Rather than relying on dedicated networks of SSA satellites as an external data provider, monitoring its own environment also provides data security for small satellites.

Observing RSOs while also providing AD is a challenging mission concept. First and foremost, both attitude determination and RSO detection require extensive computational resources. While recent studies have proposed several star tracker algorithms that are more computationally efficient and suitable for use onboard spacecraft [11,12], most algorithms are not compatible with the computing limitations of smaller (e.g., CubeSat or nanosatellites) spacecraft. Traditionally, star trackers have been too large, expensive, and power intensive for a CubeSat platform [13]. Added to this already challenging task, RSO detection is equally complex and computationally demanding. Most RSO detection relies on AI-based algorithms [4,14,15] that are not suitable for real-time operation. Multi-functional, small form-factor STs are not only difficult to develop but require careful planning in mission operation to avoid overloading power, onboard computers, and other spacecraft-level resources.

Both AD and SSA algorithms are complex on their own. Together, two sets of seemingly independent algorithms must be designed simultaneously to operate in real-time and communicate with multiple subsystems onboard a satellite. For example, if the same set of images are used for both functions, an image buffer needs to be designed to share the real-time images to process them for both functions. If a different set of camera settings are required for AD and SSA, real-time communication with the camera control software needs to be implemented, as well as power, communication, and thermal management for safe keeping and operational management.

Lastly, as described in [16], star tracker specifications required to achieve a given attitude accuracy are stringent in terms of the signal-to-noise ratio, focal length, pixel size and much more. RSO detection has slightly different, yet equally demanding requirements for camera parameters. While most are compatible (minimum detectability of dim objects), some pose competing constraints. For example, to reliably detect RSOs and have a reasonable update rate for AD, star trackers require relatively short exposure time while RSO detection requires longer exposure time to capture moving objects within the same frame.

### 1.2. Research Overview

In this paper, we present a concept of a dual-purpose camera suitable for CubeSat-class spacecraft as a star tracker capable of RSO detection. The prototype camera payload was flown on the STRATOS balloon platform in 2023 as a technology demonstration for a star tracker concept. In preparation of the balloon campaign, a series of ground-based campaigns were conducted to characterize the camera and collect night sky images for algorithm design. Ground observation data are then compared to the data collected from the stratosphere where no atmospheric interference is expected. The final balloon payload was also extensively tested in a space-like environment with thermal, vacuum, and functional tests. The primary objectives of this study, therefore, are (1) to demonstrate a low-cost camera for a CubeSat-like mission in a space-like environment; (2) to conduct real-time attitude determination and RSO detection and lastly (3) to collect night sky images from a near-space platform for future development of AD and SSA algorithm design.

## 2. Dual-Purpose Camera Technology Demonstration Payload—STARDUST

STARDUST (Star Tracker Attitude and RSO Detection for Unified Space Technologies) was a primary payload onboard the RSONAR II (Resident Space Object Near-space Astrometric Research) stratospheric balloon mission flown in August 2023 onboard the CSA STRATOS gondola.

### 2.1. Hardware Description

The hardware selected was for a star tracker-like Commercial off-the-shelf (COTS) camera system with a CubeSat form factor. The design prioritized low cost and functionality such as a WFOV lens and the sensor’s limiting magnitude. There are competing requirements between the star tracker and RSO detection functions, including the fast integration time required for the ST, compared to the longer exposure time for RSO detection to guarantee positive detection of dimmer RSOs. STs such as Rocket Lab’s ST-16RT2 [17] and Ball Aerospace’s CT-2020 [18] provide arc-second level accuracy at 2–5 Hz and 10 Hz, respectively. This comes with a higher price tag and smaller FOV (less than 15 degrees), which does not meet the STARDUST payload’s requirements.

For this project, the Raspberry Pi High Quality (HQ) Camera [19], IDS UI-3370CP-M-GL [20] and Alvium 1500 C-500 m [21] cameras were considered as the commercial-grade star-tracker like cameras. A trade study between these cameras looked at cost, temperature rating, connector types, maximum resolution, exposure time, quantum efficiency, flexibility in camera parameters, and ease of programming. It was determined the IDS UI-3370CP-M-GL camera would be suitable for the project, having no major drawbacks for operation with the benefit of larger pixel scale for low-light conditions. This is at the expense of marginally larger power, mass, and cost budgets. The IDS camera is a monochrome 1″ sensor, with a 5.5 µm pixel size, maximum resolution of 2048 by 2048 pixels, exposure time of 0.038–500 ms, and power consumption of 1.8–3.6 W. To trade off the selection of the camera’s costs, a much cheaper lens was selected for operation. The 16 mm telephoto lens, 6 mm wide-angle lens, and 25 mm 5 MP Lens for the HQ camera were all considered. The 16 mm telephoto lens was ultimately selected for the FOV of approximately 40 degrees and large aperture. Finally, the Raspberry Pi 4 Model B was selected for the onboard computer (OBC). The Raspberry Pi Zero 2 W was initially considered, but due to the computations required for a dual-purpose star tracker, a trade off was made at the cost of larger power, mass, and cost budgets. The Raspberry Pi 4 Model B uses the Broadcom BCM2711, quad core Cortex-A72 64 bit processor, 8GB LPDDR4-3200 SDRAM, and has a power consumption of 2.7–6.4 W. Figure 2 shows the selected camera and OBC.

In addition to the payload and OBC, a Power Distribution Unit (PDU) and temperature sensors were also implemented. The PDU, providing power to STARDUST and three other payloads onboard RSONAR II, was primarily used to down regulate the 36–24 V DC voltage supplied by the CSA’s gondola batteries to the 5 V input of the Raspberry Pi. This is achieved using the PYBE30-Q24-S12-T DC-to-DC converter, and two PYBE30-Q24-S5-T converters. All three regulators were placed in the application and Electromagnetic compatibility (EMC) circuits recommended by the manufacturer described in [22]. The PDU connects to the gondola batteries with a power harness using the PT06E-12-3P(SR), PT06E-12-3S(SR), and PT02E-12-3P connectors following the MIL-DTL-26482 standard and a 16 AWG wire dual-conductor wire providing a larger ampacity for the expected power draw. The operational nominal power draw for STARDUST was 8.12 W. Two K-type thermocouples with MAX31855 amplifier breakout boards were selected as the temperature sensors to monitor the IDS camera chassis. Serial Peripheral Interface (SPI) was used to measure the in-flight temperature alongside the Raspberry Pi internal sensor.

The selected hardware determined to meet the minimal requirements for the project, was then tested in-field to quantify star and RSO capture described in Section 5. The selected hardware was tested against the Raspberry Pi HQ camera with the same 16 mm lens using each respective proprietary software. The IDS camera was demonstrated to observe at least 3 magnitude stars at 100–500 ms exposure time. The HQ camera requires a minimum of 30 s exposure time to be able to observe the same stars. This would not be feasible during operation, as the gondola platform will sway and cause stars and RSOs to streak in the image leading to degradation or loss of scientific information. This was used as a validation for the selected hardware to move forward with full software and algorithm development. The dual-purpose star tracker hardware aimed to be simply implemented for future CubeSat applications, with the ability to use open-source systems for future developments.

### 2.2. Software Description

The software of the STARDUST payload was designed to initiate autonomously on power on. After ensuring that no unplanned power cycles occurred, the camera and code were initialized, performing functions such as setting the camera’s parameters and initiating counters.

The code then captured an image, extracted the centroids of the objects in those images, and saved the health and temperature sensor data, for two iterations. On the third iteration, the Lost-in-Space (LIS) attitude determination function was executed first, followed by the capturing of the third image and centroid extraction. With three images loaded into memory now, RSO detection was possible and was performed with the RSO detection algorithm. Afterwards, the time elapsed since the LIS function, as well as its return type (successful or unsuccessful) were used to determine which attitude function to use. If the time elapsed since the LIS function was greater than five minutes, or if the function did not execute successfully, the function was executed again. Otherwise, the tracking attitude determination function was executed. Regardless of the AD function used, the health and temperature sensor data were stored again before the next iteration. The next subsections serve to further explain each of the functions, with additional details for the algorithms provided in Section 3 and Section 4. Figure 3 provides a summary of the software architecture.

Initialize Camera Function:

This function was used to set the camera’s parameters, including the resolution (2048 by 2048 pixels), exposure time (100 ms), gamma (1.0), gain (100), and pixel clock frequency (100 MHz). This function also recorded the boot-up time and ensured that a power outage did not occur. If a power outage did occur, this function forced a software reboot to ensure that the entire system resets properly, avoiding issues such as driver errors.

Capture Image Function:

This function captured an image with the camera, converted the data into an array, logged the image name using the onboard date, time, and counter, saved the image with these details, and incremented the counter.

Extract Centroids Function:

This function was the first step in the RSO detection algorithm. It extracted the centroids of all the objects (stars, RSOs, noise) in an image, and converted them to x and y coordinate pairs to be analyzed by the next step of the RSO detection algorithm.

Detect RSOs Function:

This function was the second step in the RSO detection algorithm. It took in three sets (from three sequential images) of x and y coordinate pairs and identified unique x and y coordinate pairs that satisfied the conditions to be considered an RSO. This function also served to save x and y coordinate pairs to the SD card.

Lost-In-Space Attitude Determination Function:

This function used the current iterations image array captured and completed a LIS algorithm. This includes star detection and centroiding, star identification, and attitude determination. The attitude results and time are then saved into a csv file.

Tracking Attitude Determination Function:

This function used the current and previous iterations image array captured, and the previous attitude result. Star centroiding, proximity search, and attitude determination are then calculated, and the attitude results and time are saved.

Save Health and Sensor Data Function:

This function was used to save the health data reported by the Raspberry Pi OBC (the frequency of each CPU core and the CPU temperature) as well as the temperature reported by the connected temperature sensors.

## 3. Star Tracker Attitude Determination

A star tracker can provide inertial pointing with the highest accuracy relative to other AD sensors [23,24]. It is capable of having two functions during its operation; the LIS and tracking modes. LIS uses the knowledge of star positions in an inertial reference frame from known star catalogs and matches the sensor’s star measurement to find the host’s attitude. Therefore, it does not require prior knowledge of the host’s attitude. The tracking mode can update the attitude based only on the sensor measurements of the stars. This is useful to reduce computational resources following the accurate LIS execution. Both were developed for the STARDUST payload and are described below.

### 3.1. The Lost-in-Space Mode

The first step in the algorithm is to detect and centroid the stars in the image. For star detection, a binary threshold is applied followed by contour-detection. The star objects are then centroiding using the Centre of Mass (COM) method. The COM calculation considers the 3 × 3 pixel neighborhood around the estimated center and provides the sub-pixel coordinates using a weighted average of pixel coordinates based on pixel brightness. Next, the centroided stars are ordered by brightest, and the three-brightest stars are used in a pattern matching algorithm to find their corresponding inertial measurements in the Earth-Centered Inertial (ECI) frame. The planar area and polar moment method, detailed in [25], was used for pattern matching. Once the three brightest stars are selected, they are projected from the imager to the celestial sphere following [26]. Using these unit vectors ($V_{1}$, $V_{2},$ and $V_{3}$), the area and polar moment is calculated.
(1)$$Area=\sqrt{s(s-a)(s-b)(s-c)},$$
(2)$$a=\left|V_{1}-V_{2}\right|,b=\left|V_{2}-V_{3}\right|,c=\left|V_{1}-V_{3}\right|,\;s=\frac{1}{2}\left(a+b+c\right)\;Moment=Area*\left(\frac{a^{2}+b^{2}+c^{2}}{36}\right)$$

An onboard catalog was created using the Bright Star 5 catalog [27], thresholded at magnitude 3 stars and brighter, with a window of 300 to 60 degrees Right Ascension (RA) and −30 to 60 degrees Declination (DEC). A star brightness threshold can be justified given the camera’s limiting magnitude, to ensure at least three stars are in the image for most of the flight (analysis in Section 6). The RA and DEC window used was based on the planned flight and pointing of the gondola. To compare with the onboard catalog, a weighting of 80% for the Area and 20% for the Moment was used based on ground experimentation.

Finally, with the three-star vectors in the body and ECI frame, the Quaternion Estimator (QUEST) algorithm was used [28]. QUEST can rapidly find the three-axis attitude estimation, in the form of a quaternion, and has been used on hundreds of space missions to date.
(3)$$q=\frac{1}{\sqrt{\gamma^{2}+{\left|\chi \right|}^{2}}},$$
$$\gamma =\alpha \left(\lambda_{max}+\sigma \right)-det(S),\;\chi =[\alpha I_{3x3}+\left(\lambda_{max}-\sigma \right)S+S^{2}]z,\;\alpha =\lambda_{max}^{2}-\sigma^{2}+trace\left(adj\left(S\right)\right),\;\lambda_{max}\equiv found\;in\;Newton-Raphson\;method,\;\;\sigma =trace\left(B\right),\;S=B+B^{T},\;Z=\left[-B_{32},\;B_{31}-B_{13},\;B_{12}-B_{21}\right],\;B\equiv Attitude\;profile\;matrix$$

The attitude quaternion was then converted to Euler angles, specifically the Tait-Bryan angles, simplifying the mission’s post-analysis.

If two or fewer bright stars were detected, the algorithm would be re-run on the next image capture. The first successful attitude output would be saved onboard, and in the subsequent LIS iteration compared with the next successful execution. If a larger than 60-degree total angular distance between iterations is calculated, the previous attitude was used. However, this would only be completed once in a row and the next attitude would be trusted as the correct value, to avoid an initial incorrect attitude propagating throughout the mission. This provided some robustness to star misidentification, caused by an incorrect pattern match or if a bright RSO was detected in the star selection.

### 3.2. The Tracking Mode

The tracking mode requires the time between the images captured and the previous attitude to be known. First, a histogram equalization or a gaussian blur is applied to the image to help with noise reduction. Next, a gamma correction of 15 is applied to increase the brightness of the stars. At this point, the image is all black with white contours. To minimize the amount of false positive stars, a contour is only recorded if it spans an area greater than 5 pixels. The centroid of each recorded contour is then saved and compared with the star locations from the previous image using a nearest neighbor search. If the centroid of a contour is found within a predefined radius of 15 pixels of a star from the previous image, it is assumed it to be the same star. This radius was selected due to the planned inertial pointing constraints of the mission, with a small slew rate between images.

The algorithm can estimate the new attitude with just one star’s location, but it is constrained to prioritize accuracy, requiring a minimum of three star locations to be known. Each recorded star can then be represented by a unit vector with respect to the boresight of the camera. The body vectors from the current and previous images were used to find the angular rate between the two images. The time interval is relatively small between images, therefore, the assumption of a constant angular rate between images is made. The methodology from [29] was used in the implementation of the tracking mode algorithm. To find the angular rate between the two images, the body vectors of the current image and the previous image are utilized in Equation (4) below.
(4)$$\vec{\omega }=\left[\begin{array}{c} {\vec{\omega }}_{x}\\ {\vec{\omega }}_{y}\\ {\vec{\omega }}_{z} \end{array}\right]=\frac{1}{dt}{\left(\sum_{i=1}^{n}{\left[b{\left(t\right)}_{i}\times \right]}^{T}\left[b{\left(t\right)}_{i}\times \right]\right)}^{-1}\left(\sum_{i=1}^{n}{\left[b{\left(t-dt\right)}_{i}\times \right]}^{T}\left[b{\left(t\right)}_{i}\times \right]\right)$$

With the angular rate between the two images found, the new attitude can then be calculated. The attitude is found as a quaternion with Equation (5) [30]. Here, the angular velocity found in Equation (4) is represented as a pure quaternion, such that a Hamilton product can be applied. It is then converted to the omega operator, Ω, defined as the skew-matrix form of the angular velocity. Finally, the new attitude, *q*, can be found by multiplying the previous attitude, $q_{0},$ by the quaternion representation of the rotation between the two images. This algorithm is repeated until the 5 min interval of image capture is complete and the LIS algorithm is performed once again.
(5)$$q=\left[\cos\left(\left|\Omega \right|\frac{dt}{2}\;\right)I_{4}+\frac{1}{2\left|\Omega \right|}\sin\left(\left|\Omega \right|\frac{dt}{2}\right)\Omega \right]q_{0},\;q=\left[\begin{array}{c} q_{x}\\ q_{y}\\ q_{z}\\ q_{w} \end{array}\right],\;\Omega =\left[\begin{array}{c} 0\\ \omega_{x}\\ \omega_{y}\\ \omega_{z} \end{array}\;\begin{array}{c} -\omega_{x}\\ 0\\ {-\omega }_{z}\\ \omega_{y\;} \end{array}\;\begin{array}{c} -\omega_{y}\\ \omega_{z}\\ 0\\ {-\omega }_{x} \end{array}\;\begin{array}{c} -\omega_{z}\\ -\omega_{y}\\ \omega_{x}\\ 0\; \end{array}\;\right]\;$$

## 4. RSO Detection

The RSO detection algorithm was developed to detect RSOs in real time in a rolling window of three starfield images received from the camera. Several simplifications and assumptions were made when designing the algorithm. Firstly, it was assumed that the background stars captured in the images would not appear to move more than one pixel between each image (the movement of which would primarily be from the gondola’s sway). Next, it was assumed that RSOs would appear to be travelling mostly linearly across the FOV of the imager. Finally, it was assumed that RSOs would travel the same amount of distance across the FOV of the imager, between each image that captured the RSO (equidistance).

The algorithm works in two main steps as described in Section 2.2. Software Architecture. The first step consists of extracting the centroids of the objects in each image, while the second step consists of detecting RSOs across three sets of these centroids.

### 4.1. Extracting Centroids

In the first step, the algorithm begins by binarizing the image by applying a simple threshold to the image. Connected Component Analysis (CCA) using 8-pixel-neighbourhood connectivity is then performed to uniquely segment each of the objects in an image [31]. This algorithm returns details for each segmented object, most notably the x and y sub-pixel locations of each object’s centroid and the pixel size of each object. This list of centroids is then filtered to remove objects deemed too small or large to be considered stars or RSOs, such as illumination effects or hot pixels. The size of this point list (which corresponds to the number of detected objects) is then analyzed to determine an appropriate threshold to use for the next iteration. If there are too few objects, the threshold is reduced, thereby allowing dimmer objects to be picked up by the algorithm. If there are too many objects, the threshold is increased, having the opposite effect. This point list (and two more-point lists, corresponding to a sequence of three images) is passed on to the next step of the algorithm. Figure 4 below shows the block diagram outlining the steps of the extracting centroids step.

### 4.2. Detecting RSOs

In the next step, after ensuring that the three point lists from the previous step contain data, the three point lists are looped through. For each set of unique (one point from each point set), unmatched set of three points, the Euclidean distance is calculated between the point belonging to the first image and second image, d1, and again calculated between the second image and third image, $d_{2}$. These distances are then checked to ensure they are far enough to be considered RSOs, as defined in the assumptions mentioned previously. Equation (6) below is then used to determine how similar these distances, followed by Equation (7) to determine the angle between the vectors, θ (where $\vec{d_{1}}$ is the vector from point one to point two, and $\vec{d_{2}}$ is the vector from point two to point three). These two calculations are performed to fulfill the linear and equidistant RSO assumption.
(6)$$d_{similarity}=1-\left|\frac{d_{1}-d_{2}}{max(d_{1},d_{2})}\right|$$
(7)$$\theta =\cos^{-1}\left(\frac{\vec{d_{1}}\cdot \vec{d_{2}}}{\left|\vec{d_{1}}\right|\left|\vec{d_{2}}\right|}\right)$$

From experimentation with using this formula on existing optical RSO imagery, it was found that the angle was related to the distance similarity by Equation (8) below. This equation also provides the maximum angle, and the angle is deemed small enough to consider the triplet of three points as a roughly linearly moving RSO.
(8)$$\theta_{max}=39d_{similarity}-8$$

The triplet of three points is then marked as matched, and the next set of points are analyzed until all points have been considered. The function then saves these RSO points to the OBC’s SD card. Figure 5 below shows the block diagram outlining the steps of the detecting RSOs step.

## 5. Image Collection

### 5.1. Field Campaigns

Several ground-based field campaigns were conducted to optimize camera parameters including gain, exposure time, image resolution, and gamma. Permutations of the above parameters were tested while taking images of the night sky with stars and RSOs present. These images were used to test and fine-tune the AD and RSO detection algorithms.

King City, Ontario was the first field campaign location the payload was taken to for night-sky observations. Based on experimentation, an exposure time of 100–200 ms, image resolution of 2048 by 2048 pixels, and gamma of 1 was selected. The key parameter, exposure time, was proven to see bright stars of three magnitude and multiple RSOs within the campaign. It was also shown that the Raspberry Pi HQ camera was only able to see these same objects at a much larger exposure time of 30 s. This would not be possible during the stratospheric balloon campaign, due to the instability of the gondola causing streaking of objects at larger exposure times. Without a full quantification analysis of how exposure times on the gondola affect streaking, the lowest exposure time that was able to see bright stars and RSOs on the ground was the primary variable to minimize. Finally, the payload was tested in Timmins, Ontario, and the selected parameters were verified similar to King City. Here, the lighting conditions were the best of the two locations due to minimal light pollution.

Through these ground field campaigns and algorithm tests, it was verified that the barrel distortion caused by the camera lens would negatively affect the results of the AD. MATLAB’s 2023a Camera Calibration Toolbox was used for the calibration and the algorithms were re-tested in Timmins, Ontario. In addition, the adaptive thresholding used in the RSO detection algorithm was verified during the multiple field campaigns. Here, we noted that the observing location, time, and pointing direction would vary the imaging conditions. Consequently, the initial threshold value would adjust to detect potential RSOs and stars within the images captured.

### 5.2. STRATOS Campaign

The STARDUST payload was flown on the CSA/CNES STRATOS Balloon platform on 22 August 2023, from 4:52 am to 9:24 am Coordinated Universal Time (UTC). During the 4.5 h flight, a total of 23,354 images were collected from the STARDUST payload. However, the gondola reached the ceiling of 37 km and was stabilized after approximately 3 h. This meant that the scientific data captured were only for 92 min. This period was ideal for RSO viewing time, which is approximately 2 h before local sunrise or after local sundown. The flight profile once stabilized was inertial pointing, which meant the camera was staring about a specific section of the sky. In this case, the camera boresight was pointing at approximately 355 degrees RA and −10 degrees Declination Dec in the celestial sphere. The mission successfully captured images and saved its AD and RSO detection results and health status onboard, for further analysis on the ground. Figure 6 below shows an image with stars and an RSO during operation.

## 6. Results

The total flight was 4 h and 32 min, from gondola lift-off to payload shut down pre-descent. During the first approximately 3 h ascent and gondola stabilization period, the attitude and RSO detections algorithms were not accurate due to the slewing of the gondola. This led to the stars streaking in the image or moving drastically from frame to frame. Therefore, the analysis shown is during the final 92 min period of inertial pointing. During this inertial pointing stage, there was some oscillating gondola motion which may have negatively impacted the results. Furthermore, the flight duration was significantly shorter than expected, with an initially planned post-sunset and pre-sunrise observation, decreasing the potential analysis window for RSO detection. Together, with the 100 ms image capture and both the real-time attitude and RSO detection algorithms, there was an average of 502 ms and standard deviation of 66 ms between each iteration. This shows potential in real-time applications for a dual-purpose star tracker, which would require a 1–10 Hz sampling rate.

### 6.1. Real-Time Attitude Determination

We had a total of 19 LIS and 11,068 tracking mode calculations. Both attitude algorithms used the open source astrometry.net application as its pointing truth data, which reports a 99.9% success rate [32]. The absolute error for each is presented and discussed below.

The LIS cross boresight error was 36.72 arc minutes, and around boresight error was 65.98 arcminutes. The star identification stage had an 84% true positive rate, in which the previous attitude was used as a substitution. Figure 7 shows the AD LIS in-flight results.

Upon further inspection during post-mission analysis, it is likely that the barrel distortion was causing the largest error during the attitude determination QUEST algorithm. The pixel scale for the system is 71 arcsec per pixel, or approximately 1.2 arcmin per pixel, so there was a preliminary expectation that the final accuracy would be on the order of arc minutes. Improvement of lens selection can benefit the final attitude accuracy to this end. Similarly, a star tracker algorithm using four or more stars in the algorithm can increase the accuracy at the cost of computational speed and resources.

Using a simplified expected pointing direction accuracy for star trackers [33], ${PD}_{estimated}$, the estimated accuracy of the system was 3.4 arcminutes. 103 stars in the onboard catalog, $N_{Catalog}$, was dependant on the magnitude limit of the sensor while maintaining at least three the stars in the FOV, $N_{FOV}$. The average hyperacuity of centroiding accuracy of 0.1 pixels, $E_{Centroid}$, was selected based on the previous ground campaigns and testing using the custom star field simulator (more description of the simulator in [34]). The primary contribution to the centroiding error was due to the selection of the imagers bit depth of 8-bit. Increasing the bit-depth to the imager’s maximum 12 bit would provide additional pixel intensity information and therefore increase the estimate of the sub-pixel coordinates using the COM method.
(9)$$N_{FOV}=\frac{N_{Catalog}-N_{Catalog}*\cos\left(\frac{FOV}{2}\right)}{2}=3.17\;stars$$
(10)$${PD}_{estimated}=\frac{E_{Centroid}}{\sqrt{N_{FOV}}}=0.056\;\left[deg\right]=3.4\;[arcmin]$$

The second image in the sequence was the first star misidentification. It was noted in the post-analysis that one of the stars selected was near the edge of the image border. Here, being the furthest from the principal point, the largest effects of barrel distortion caused a misidentification of a three-star group in the area and moment calculation. This was noticed in the ground campaign and led to the weighting of 80/20 between area and moment, respectively, to partially mitigate these effects. The thirteenth and last images contained an RSO, which also led to an incorrect star identification and therefore incorrect attitude. It would be ideal to merge the RSO detection and attitude determination code, to remove RSO’s from the sequence and further improve the true positive rate.

This level of accuracy can be suitable for nanosatellite applications that require a degree of accuracy for science or engineering. Regarding SSA, this accuracy would be required to be further improved for algorithms such as RSO identification.

The tracking mode cross boresight error was 1.38 degrees, and around boresight error was 2 degrees. The tracking mode was able to accurately update the attitude during the 5 min intervals before returning to the LIS mode within an error of 10 degrees excluding anomalies. Preliminary analysis has shown a decrease in computation time upwards of 100 ms compared to the LIS algorithm.

Figure 8 shows the attitude errors during the inertial pointing period for the tracking mode. As seen in Figure 8 at the 25 and 75 min mark (1500 and 4500 s), there is a large spike in inaccuracy. These spikes in inaccuracy are likely caused by stars being incorrectly identified. If there are multiple stars or RSOs being detected within the search radius of the known star in the previous image, the algorithm is unable to determine which star is the same as the previous one. This can cause the calculated centroid of the star to vastly differ from the true values. To mitigate inaccuracy in the detection stage, an Iterative Closest-Point (ICP) algorithm can be considered for implementation to replace the current algorithm.

### 6.2. Real-Time RSO Detection

Given that the mission was a technology demonstration, the main objective for the RSO detection segment of the payload was to detect any RSOs at all, during the flight. Success would be achieved if an RSO detection reported by the algorithm’s output was verified visually against the corresponding raw images. Given this objective, these results do not consider metrics such as precision and recall, and instead seek to quantify and verify the number of RSOs reported by the algorithm and the number of total detections corresponding to each RSO. To give an understanding of the consistency of the detections, the longest consistent detection of each RSO is also given. These results are provided in Table 1.

By plotting the pixel centroids of an RSO, corresponding to each of the three sequential images it was observed in, a visual was constructed to represent how the RSO detection algorithm works, and what it outputs. Figure 9 below is an example of this.

Upon analyzing the raw images captured by the STARDUST imager, it was observed that the lit pixels corresponding to stars and RSOs in the images were fewer and dimmer (in terms of pixel intensity value) than what was observed during ground campaigns. The cause of this is currently being investigated, with the current hypotheses being that there was a fogging in the optics, or temperature variations throughout the mission and the harsh vacuum of the stratospheric environment resulted in degradation of the imager. Due to this suspected degradation, most of the RSOs in the images corresponded to a single pixel each, which would not be detected by the algorithm. This was a result of the algorithm imposing a minimum pixel threshold of 10, used to filter out noise, and was experimentally determined during ground campaigns. This meant that objects in the images corresponding to fewer than 10 pixels would be ignored, which includes these single-pixel RSOs. Nevertheless, the algorithm was able to detect RSOs in the images, given that there were some larger and brighter RSOs during this imaging window.

The dynamic threshold function implemented in the algorithm proved to be very useful, given that the algorithm automatically decreased the threshold to account for the fewer points it was detecting during the mission. This allowed fainter RSOs to be detected, though they had to pass the minimum pixel threshold as well. The algorithm could certainly benefit from better object analysis in the images, employing a dynamic analysis of the points similar to the dynamic threshold. For example, when fewer than desirable points are detected in an image, the algorithm could reduce the minimum pixel threshold to allow more points to be considered.

The linear motion model appears to be a good estimate of RSO motion in these images, given that the RSOs that passed the threshold and pixel area requirement were consistently detected in the images, as the RSOs passed through the FOV of the imager. The built-in tolerance to the linear motion model as defined in Equation (8) proved to be invaluable in detecting RSOs with the gondola movement, since this movement also caused RSOs to move non-linearly in the images. The equidistant requirement for RSO detection, as enforced by Equation (6), was also a good estimate of the motion of the RSOs.

Though intended to be a technology demonstration, the images captured by the STARDUST payload have proved to be an invaluable dataset for SSA, given the presence of numerous RSOs. Furthermore, a unique opportunity is offered by the challenges with the dataset, including the drastic background star movement (caused by the gondola movement) and the faint, small RSOs.

## 7. Conclusions and Future Work

The STARDUST stratospheric balloon payload successfully performed real-time AD and RSO detection during the inertial pointing phase of the STRATOS 2023 campaign. In this paper, a low-cost, WFOV dual-purpose star tracker was demonstrated as a technology that can augment SSA for stratospheric balloons and space-based applications.

The AD results showed sub-degree accuracy in the LIS mode and under five-degree accuracy in the tracking mode. The star identification, attitude determination, and star tracking were negatively affected by barrel distortions produced by the lens. This can be mitigated to some extent with improved lens selection and camera calibration. The star centroiding, star identification, and attitude determination algorithms selected can be further compared to its more accurate and computationally costly algorithms such as the Gaussian curve fitting method, pyramid methods, and Extended Kalman Filter (EKF), respectively. In addition, while the AD and RSO detection and algorithms were isolated to reduce risk, future iterations of the mission will seek to combine the algorithms to improve their respective performance. For example, the RSO detection algorithm could be used to remove RSOs in the images that are passed to the attitude determination algorithm. This would be helpful in reducing false detections in the attitude determination algorithm from the RSOs present in the images.

The real-time RSO detection algorithm was successfully able to detect RSOs during the mission. Though the RSOs within the image appeared much smaller and fainter than what was observed during ground campaigns, the algorithm was able to detect 11 unique RSOs corresponding to 669 total detections in the 92 min analysis window. The dynamic image thresholding technique and linear motion model proved to be excellent algorithms in consistently capturing the RSOs that passed the minimum pixel threshold. However, it was determined that the minimum pixel threshold itself needs improvement, given that many RSOs observed in the raw images were much smaller than the 10-pixel threshold.

Several improvements could be made to the RSO detection algorithm and are planned to be incorporated in future research. As mentioned, the swaying of the gondola, which caused the background stars to appear to move throughout the images, suggests that a method to correct the apparent motion of the stars could be added to create a more robust RSO detection algorithm. Such a method is currently being developed and is being tested with optical imagery from the Fast Auroral Imager (FAI) onboard the CASSIOPE satellite. Another improvement to be made is the addition of a tracking method to give each detected RSO a unique identity, carried through its detection in subsequent images. This can be used to identify RSOs of particular interest, in real time. Such a tracking implementation is also being investigated for future work. Lastly, the RSO detection algorithm can be improved by incorporating a more advanced motion model than what was experimentally determined. While the linear motion model appears to work effectively for many of the RSOs found in the images captured from the stratosphere, RSOs in general do not always appear to be moving linearly through an imager’s FOV.

## Figures and Tables

**Figure 1 sensors-24-00071-f001:**
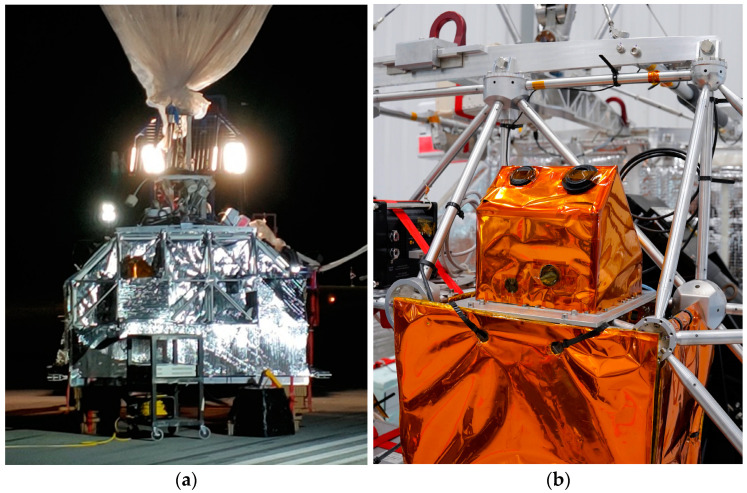
Star tracker payload on 2022 STRATOS Balloon Platform (**a**), and 2023 STRATOS Balloon Platform (**b**).

**Figure 2 sensors-24-00071-f002:**
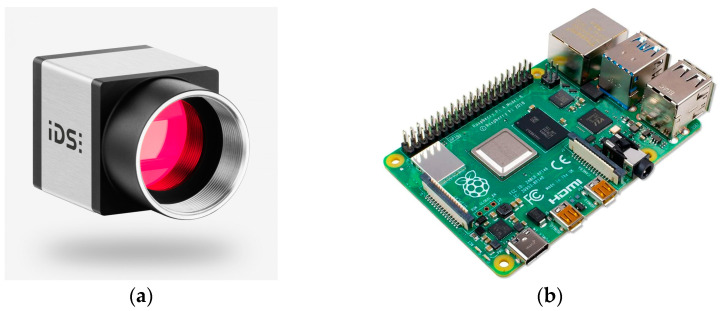
IDS UI-3370CP-M-GL camera (**a**), and Raspberry Pi 4 Model B OBC (**b**) used on STARDUST.

**Figure 3 sensors-24-00071-f003:**
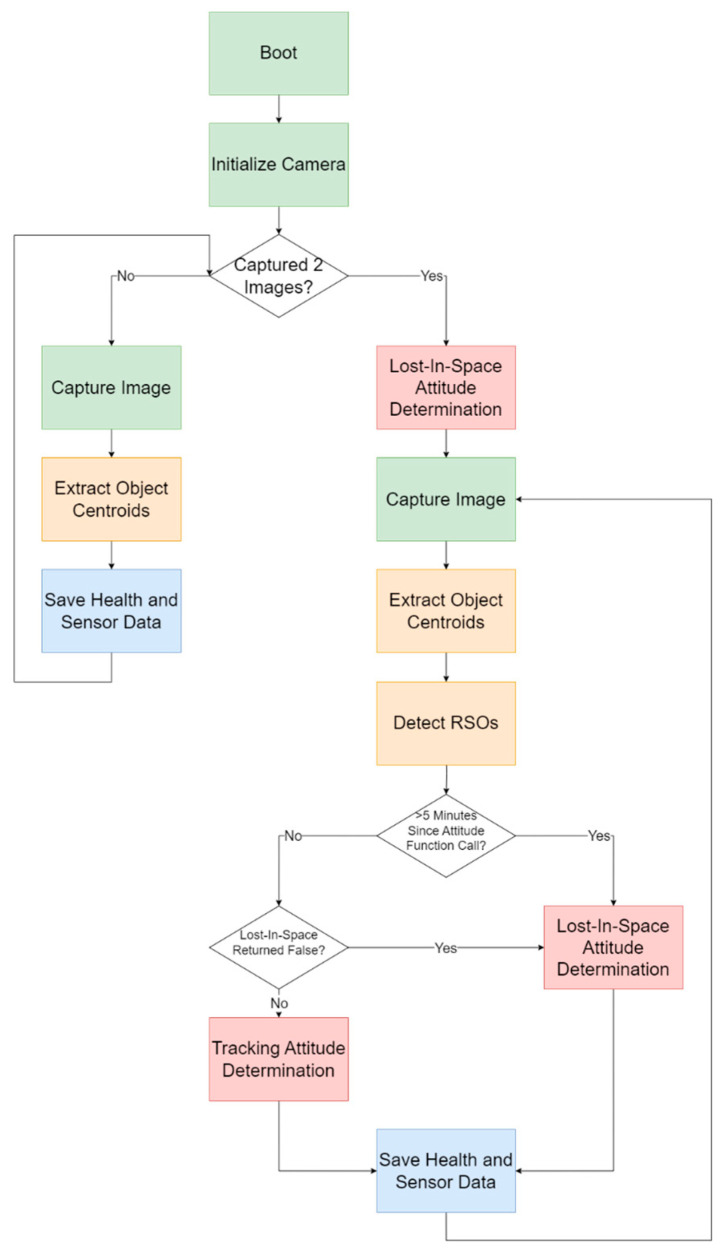
Software block diagram for STARDUST, visually outlining the functions and logic.

**Figure 4 sensors-24-00071-f004:**
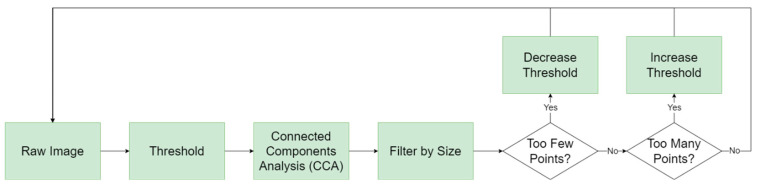
Block diagram outlining the first step, extracting centroids, in the RSO detection algorithm.

**Figure 5 sensors-24-00071-f005:**
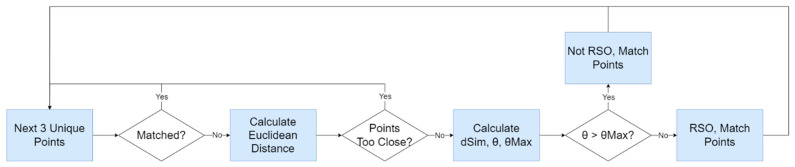
Block diagram outlining the second step, detecting RSOs, in the RSO detection algorithm.

**Figure 6 sensors-24-00071-f006:**
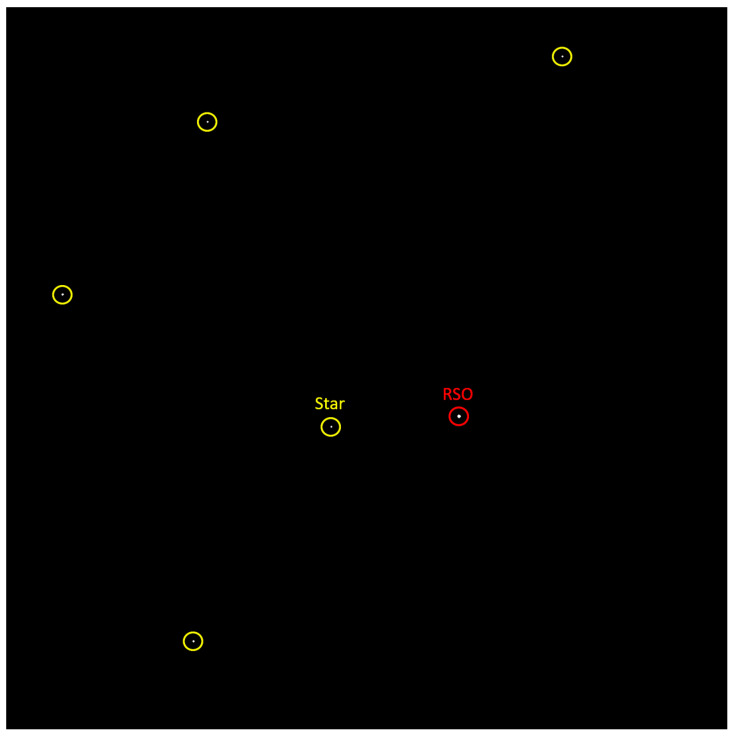
Processed starfield image with an RSO captured during the STRATOS 2023 campaign.

**Figure 7 sensors-24-00071-f007:**
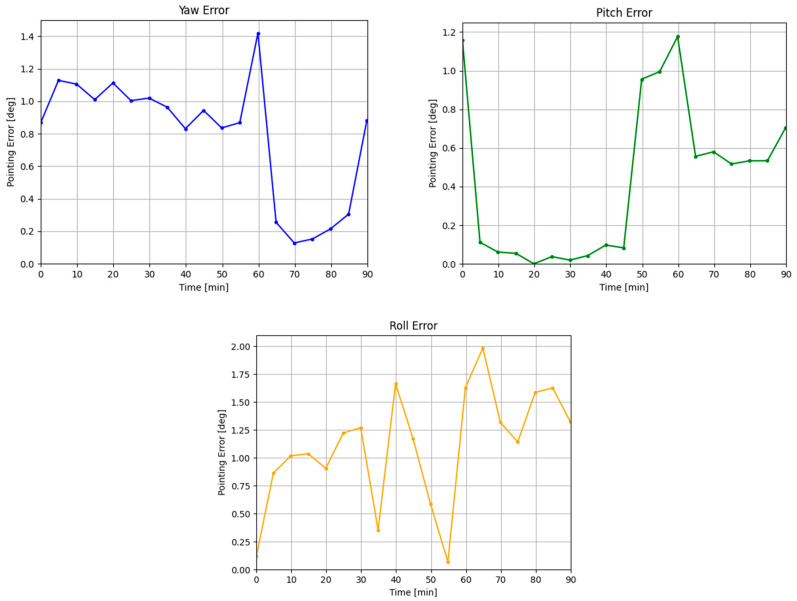
Attitude pointing Yaw, Pitch, and Roll errors for the LIS algorithm.

**Figure 8 sensors-24-00071-f008:**
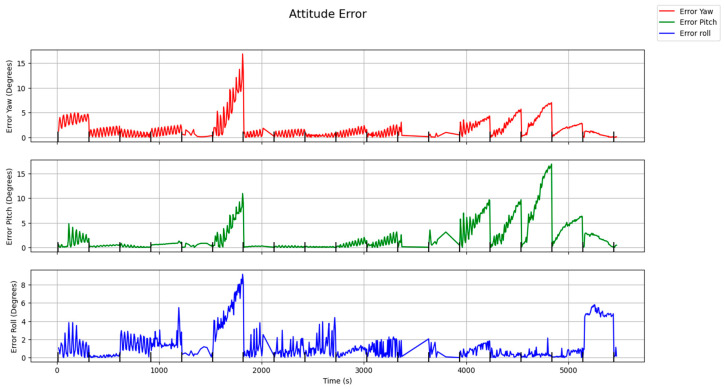
Attitude pointing Yaw, Pitch, and Roll errors for the tracking mode algorithm.

**Figure 9 sensors-24-00071-f009:**
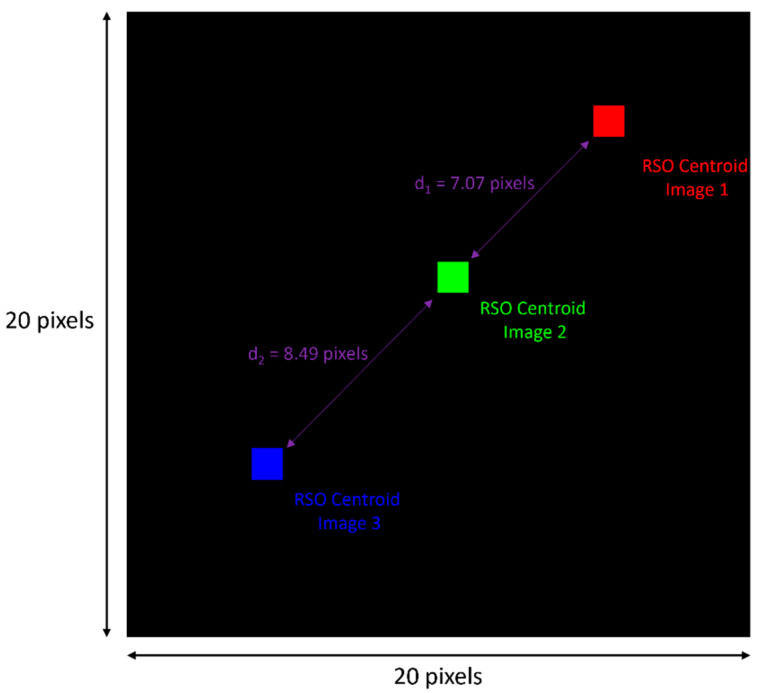
RSO centroids corresponding to three sequential images, plotted as single, colored pixels on a black background. The Euclidean distances calculated by the algorithm are also plotted.

**Table 1 sensors-24-00071-t001:** RSO detection results, with total detections and longest consecutive detection included.

RSO Number	Total Detections [Images]	Longest Consecutive Detection [Images]
1	53	18
2	34	34
3	40	40
4	36	36
5	14	14
6	66	55
7	31	16
8	106	106
9	13	6
10	159	159
11	117	117
Total RSOs: 11	Total Detections: 669	

## Data Availability

The data presented in this study are available on request from the corresponding author. The data are not publicly available due to continuing research and containing sensitive data.

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
