# Peer review of "A Dual-Purpose Camera for Attitude Determination and Resident Space Object Detection on a Stratospheric Balloon"

_sensors, 2023, doi:10.3390/s24010071_

Round 1
Reviewer 1 Report
Comments and Suggestions for Authors
Please see my comments in the attached file review_v01.pdf.

Comments on the Quality of English LanguageIn general it is good, but some sentences are incomplete.
Reviewer 2 Report
Comments and Suggestions for Authors
Dear Author,
I find your paper to be an excellent exploration of a novel dual-purpose camera, proficient in determining both attitude and space objects.
The focus on real-time data processing algorithms and hardware is commendable and makes a strong case for publication. However, as someone engaged in space target detection, I'm particularly intrigued by the centroid algorithm. In our work, where cameras with larger dynamic ranges are common, we typically encounter centroid errors in the range of 0.1 to 0.01 pixels. I am curious about the centroid error in the algorithm proposed in your paper and the primary factors contributing to it. A more in-depth discussion on this aspect would add significant value.
These are the main points I wanted to convey.
Congratulations on a well-structured paper.
Cheers.
Comments on the Quality of English LanguageNo
Reviewer 3 Report
Comments and Suggestions for Authors
The abstract presents a summary of the paper but does not clearly state the main contribution of the authors.
Expand on the related works to express the motivation of this work.
Kindly provide detail on equation 5, expressing “omega”, the exact skew symmetric form of the angular velocity.
Comments on the Quality of English LanguageMinor revision on the quality of English.
Reviewer 4 Report
Comments and Suggestions for Authors
(1) In the Introduction section, the necessity of implementing both AD and SSA algorithms using cameras is not detailed enough.It can be described in more detail.
(2)In the process of RSO detection, the points with fewer pixels were ignored and the threshold was adjusted.The adjustment algorithm needed more verification to ensure its rationality
(3)In the tracking mode, the angular error reaches a maximum of 10 degrees, whether the influence of such a large error on the attitude fixing accuracy exceeds the requirements, and whether the attitude determination in the tracking mode can be considered through the MEMS gyroscope.
